# Cardiac Function Evaluation after SARS-CoV-2 mRNA Vaccination in Children and Adolescents: A Prospective Speckle-Tracking Echocardiography Study

**DOI:** 10.3390/vaccines11081348

**Published:** 2023-08-09

**Authors:** Jolanda Sabatino, Costanza Di Chiara, Daria Lauretta, Jennifer Fumanelli, Greta Luana D’Ascoli, Daniele Donà, Sandra Cozzani, Andrea Oletto, Carlo Giaquinto, Giovanni Di Salvo

**Affiliations:** 1Division of Pediatric Cardiology, Department for Women’s and Children’s Health, University of Padua, 35128 Padova, Italy; daria.lauretta91@gmail.com (D.L.); jennifer.fumanelli@gmail.com (J.F.); giovanni.disalvo@unipd.it (G.D.S.); 2Paediatric Research Institute (IRP), Città Della Speranza, 35127 Padua, Italy; 3Division of Pediatric Infectious Diseases, Department for Women’s and Children’s Health, University of Padua, 35128 Padova, Italy; costanza.dichiara@phd.unipd.it (C.D.C.); daniele.dona@unipd.it (D.D.); sandracozzani@gmail.com (S.C.); carlo.giaquinto@unipd.it (C.G.); 4Penta–Child Health Research, 35127 Padua, Italy; andrea.oletto@pentafoundation.org

**Keywords:** COVID-19 vaccination, speckle-tracking echocardiography, children, mRNA vaccines, echocardiography

## Abstract

Background: Possible cardiac impairment after SARS-CoV-2 mRNA vaccination is a common driver of parental vaccine hesitancy. We performed a comprehensive echocardiographic evaluation of biventricular function in vaccinated children with or without previous COVID-19 compared to healthy controls. Methods: We conducted a single-center, prospective, case–control study enrolling children and adolescents aged 5–18 years attending the pediatric clinic of the University Hospital of Padua from April to June 2022. Three months after receiving the primary mRNA vaccination or booster dose, the patients underwent a cardiac assessment, including standard echocardiography and speckle-tracking echocardiography (STE). A pre-pandemic historical cohort of age- and gender-matched healthy children were used as a control. Results: A total of 39 post-VACCINE cases (24, 61% female), mean age 12.6 ± 2.6 years (range 8–17), were enrolled in the study. Ninety percent (N = 35) of patients were previously healthy. No differences in left ventricular diameters, left ventricular ejection fraction (LVEF), and tricuspid annular plane systolic excursion (TAPSE) were observed between cases and controls. Global longitudinal strain (GLS) was in the normal range in all individuals, with no differences between post-VACCINE cases and controls (−21.7 ± 2.3% vs. 21.2 ± 1.8%; *p* = 0.338). However, GLS was found to be slightly but significantly reduced in post-VACCINE children with a previous COVID-19 compared to naïve-vaccinated individuals (post-VACCINE+COVID-19: −19.9 ± 1.1% vs. post-VACCINE-only: −22.0 ± 2.3%; *p* = 0.002). Conclusions: We did not observe an impairment in GLS or in other indices of LV structure or function after mRNA COVID-19 vaccination.

## 1. Introduction

Children are largely spared from severe COVID-19 compared to adults [1,2]. However, after a few weeks from SARS-CoV-2 infection, they may develop a life-threatening acute complication, such as a multi-system inflammatory syndrome (MIS-C), myocarditis, and pericarditis, as well as long-term consequences, defined as a long-COVID syndrome, which can potentially affect the physical and mental health of children and their parents [3,4,5]. 

Vaccines are considered the most promising approach for containing the outbreak and reducing COVID-related morbidity and mortality worldwide. Growing evidence, in fact, demonstrates the crucial role of COVID-19 vaccination in reducing the incidence of MIS-C and myocarditis in the pediatric population [6,7]. Moreover, recent findings documented a lower risk of developing long-COVID syndrome in vaccinated adults compared to those unvaccinated [8,9]. 

However, during the ongoing SARS-CoV-2 outbreak, we are facing a simultaneous spike in the vaccine hesitancy pandemic [10], which is included among the top ten global health threats for decades [11]. Although several placebo-controlled trials documented the effectiveness and safety of COVID-19 vaccines in children and adults, the COVID-19 vaccination acceptance rates vary greatly globally [12]; in Italy, only 47% of individuals aged 12–19 years of age received the first booster dose of the COVID-19 vaccine and only 35% of uninfected children aged 5–11 years completed the two-dose primary vaccination [13].

The low perceived risk of severe COVID-19 in children, complacency, and concerns about the vaccine’s safety are the main drivers of parental vaccine hesitancy [14,15]. In particular, the misperceptions of a higher risk of myocarditis in adolescents and young adults after vaccination have been reported as factors contributing to delay in acceptance or refusal of COVID-19 vaccination [16,17]. 

However, since SARS-CoV-2 infects host cells through ACE2 receptors, which are largely expressed in the heart, the infection is supposed to lead to more risk of cardiac involvement than vaccination [18]. In fact, previous findings demonstrated that also asymptomatic and mild COVID-19 may induce both subclinical myocardial injury with left ventricular deformation in children and adolescents as in adults [19,20] up to 240 days from infection, regardless of the severity of acute disease [21]. Similarly, other authors described the reduction in global longitudinal strain and left ventricular strain in children with normal left ventricular ejection fraction and without any cardiac symptoms after MIS-C, suggesting a probable subclinical myocardial persistent injury in children after COVID-19 [22,23].

On the other hand, a crescent number of studies showed a significantly lower risk of myocarditis and pericarditis in vaccinated adolescents and adults compared to SARS-CoV-2 infected cases [24,25].

To date, data on echocardiographic evaluation after the COVID-19 vaccine is still lacking, especially in the pediatric population. 

Thus, to provide a detailed evaluation of cardiac function after SARS-CoV-2 mRNA vaccination, we herein describe the standard echocardiography and speckle-tracking echocardiography (STE) parameters in a cohort of previously healthy children and adolescents who underwent COVID-19 vaccination after or not a previous SARS-CoV-2 infection compared to healthy controls recruited at the Department of Women’s and Children’s Health of Padua University Hospital.

## 2. Materials and Methods

### 2.1. Setting

The COVID-19 Family Cluster Follow-up Clinic (CovFC) was established in March 2020 at the Department of Women’s and Children’s Health, University Hospital of Padua (Veneto region, Italy), with the aim of evaluating and describing the clinical and immunological impact of SARS-CoV-2 infection as well as the safety and effectiveness of the COVID-19 vaccination in children, older siblings, and parents who experienced a household SARS-CoV-2 infection.

The CovFC is a multidisciplinary pediatric clinic, including pediatricians with expertise in the field of pediatric infectious diseases, cardiology, pneumology, psychologists and neuropsychiatrists, virologists, and immunologists. The CovFC includes COVID-19 family clusters with at least one child in the pediatric age (≤18 years old).

At the enrollment, a pediatrician collects sociodemographic and clinical data, including the vaccinal status, of each family member. In addition, a blood sample is collected from all individuals for the serological assessment of SARS-CoV-2 infection, through the detection of the anti-receptor binding domain (RBD) antibodies against SARS-CoV-2 spike protein (MAGLUMI™2000 Plus, Snibe Diagnostics, Snibe Diagnostics, New Industries Biomedical Engineering Co., Ltd. (Snibe), Shenzhen, China) [26,27]. After the first evaluation, subjects are followed up for longitudinal clinical, cardiological, and serological evaluation at least 12 months after SARS-CoV-2 infection and/or COVID-19 vaccination.

The CovFC cohort includes approximately 480 families, for a total of 1500 pediatric and adult patients. Overall, it has individuals who got infected with SARS-CoV-2 within their family clusters and subjects who did not develop an infection, as well as both individuals who were vaccinated against COVID-19 and subjects who were not. 

The study protocol was approved by the local received an Ethics Committee (Prot. N° 0070714 of 24 November 2020; last amendment Prot. N° 0024018 del 5 April 2022). Parents or legally authorized representatives have been informed of the research proposal and provided written consent to use the routine patient-based data for research purposes. 

### 2.2. Study Design, Study Population, and Definitions

We conducted a single-center, prospective, observational, case–control study on Italian children and adolescents attending the CovFC of the University Hospital of Padua between April and June 2022. The study design is shown in Figure 1.

Three months after receiving the primary mRNA SARS-CoV-2 vaccination or booster dose, children and adolescents aged 5–17 years who were already enrolled in the CovFC, underwent a post-vaccination cardiac assessment, including standard echocardiography and speckle-tracking echocardiography (STE). 

At the same time as the evaluation, children’s parents were asked to fill in an ad hoc clinical questionnaire aiming to collect additional patient-level data, including comorbidities, cardiac past medical history, and previous echocardiography evaluations, SARS-CoV-2 infection-related data and long-term persistence of symptoms after infection, date and the number of received COVID-19 vaccine doses, and data on any short and long-term adverse events following COVID-19 vaccination. The survey was developed using the REDCap platform (Vanderbilt University, Nashville, TN, USA) hosted on the server of the University of Padua. Clinical data collected through the survey, as well as data collected during the visit, were then anonymized.

Children were considered fully vaccinated for COVID-19 and were defined as post-VACCINE cases if: (a) they were inoculated with two doses of Comirnaty-BioNTech/Pfizer or Spikevax vaccine; (b) they were inoculated with one dose of the Comirnaty-BioNTech/Pfizer or Spikevax vaccine within 12 months after a previous laboratory-confirmed COVID-19. Children older than 11 years old who received the booster dose 4 months after the primary regimen were considered “vaccinated and boosted” cases. The COVID-19 vaccine delivery timeline in the Italian pediatric population approved by the Italian Drug Agency was reported in Appendix A.

Post-VACCINE individuals were also classified in: (a) post-VACCINE+COVID-19 cases if they recorded at least one positive SARS-CoV-2 nasopharyngeal swab (NPS) and/or a positive SARS-CoV-2 serology before vaccination; (b) post-VACCINE-only cases if they had not clinical and analytical evidence of previous COVID-19. For each confirmed post-VACCINE+COVID-19 case, a baseline date of SARS-CoV-2 infection was clinically defined as follows: (1) for symptomatic cases, the first date between the onset of symptoms or the date of first positive NPS; (2) for asymptomatic cases, the date of the first positive molecular and/or antigenic assay. 

Post-VACCINE+COVID-19 cases were classified according to the predominant circulating SARS-CoV-2 variant of concern (VOC) in the Veneto Region at the time of children’s infection onset (baseline) using the CovSPECTRUM [28]. Any SARS-CoV-2 infection that occurred in the Veneto region from February 2020 to 14 June 2021 had a probability greater than 96% to be caused by Parental VOC; any SARS-CoV-2 infection that occurred in the Veneto region from 14 July 2021 to 11 December 2021 had a probability greater than 96% to be caused by Delta VOC; any SARS-CoV-2 infection that occurred in the Veneto region from 7 January 2022 to 8 December 2022 had a probability greater than 96% to be caused by Omicron VOC.

The severity of COVID-19 was assessed following the WHO classification [29].

### 2.3. Control Group

The study population was compared with an equal proportion of age- and gender-matched healthy children who were evaluated at the Department of Women’s and Children’s Health, University Hospital of Padua from April to June 2018, before the onset of the COVID-19 pandemic [30]. These children, defined as controls (CTRL), underwent echocardiography as part of a familiar screening process for congenital heart disease and were found to be otherwise healthy. Standard echocardiographic views were recorded at the time of the first clinical examination according to our reference protocol [31,32] and were then used for the actual offline research analyses, as described below.

### 2.4. Cardiac Evaluation

All the study population underwent standard echocardiographic evaluations using the GE Vivid E95 Ultrasound System (GE Healthcare, Waukesha, WI, USA).

The methodology for STE analysis was previously described [19,21]; it was performed by two experienced echocardiographers blind to the clinical data and COVID-19 vaccination status. 

Left ventricular diameters (LVEDD: left ventricular end-diastolic diameter; LVESD: left ventricular end-systolic diameter) and thickness (IVSd: interventricular septum in diastole; PWd: posterior wall in diastole) were normalized for body size and age and showed as z-scores (Boston Children’s Hospital z-score system) [31]. The modified Bernoulli equation was used to estimate the right ventricular systolic pressure (RVSP) from the maximum tricuspid regurgitation velocity.

The right ventricular function was assessed by calculating in millimeters the tricuspid annular plane systolic excursion (TAPSE).

Left ventricular ejection fraction (LVEF) was calculated by TTE using the Simpson method, and left ventricular longitudinal strain analysis (GLS: global longitudinal strain) was performed via 2D speckle-tracking echocardiography (STE) using GE EchoPac software (GE Healthcare, Waukesha, WI, USA).

The STE study consists of selecting the clearest frames of the apical echocardiographic projections of the heart using the GE EchoPac Software and examining the left ventricle in two, three, and four-chamber views. Then, three points are selected on the frames: two annular and one apical for the left ventricle. The software then semi-automatically identifies the chamber’s wall contour, which is manually corrected by carefully following the endocardial margins. By tracking the movement of the myocardium during the cardiac cycle, finally, the GE EchoPac’s algorithm calculates the Global Longitudinal Strain (GLS) of the heart’s wall segments.

### 2.5. Reproducibility

Intraclass correlation coefficient (ICC) is used to assess the reliability of measurements made by different observers or raters. ICC values range from 0 to 1, with higher values indicating greater reliability. In the context of this study, ICC was used to evaluate the agreement between the two different observers and the same observer at different time points for the measurements obtained from echocardiographic evaluations. The ICC values were calculated using the two-way mixed effects model with an absolute agreement definition. ICC values of 0.75 or higher are generally considered to indicate good reliability.

### 2.6. Statistical Analysis

Descriptive statistics were used, such as percentages and mean with standard deviation (SD), to summarize categorical and continuous variables, respectively, and were performed using the SPSS version 22.0 statistical package (SPSS Inc., Chicago, IL, USA). The Shapiro–Wilk test was employed to test for the normality of data. The Student t-test was used as a parametric test to compare the means of two groups of normally distributed continuous data. The Mann–Whitney U-test was employed as a non-parametric test to compare the medians of two groups of continuous data that are not normally distributed. The sample size was calculated to test the non-inferiority of Post-VACCINE individuals compared to non-vaccinated controls, with GLS as the continuous outcome, as described elsewhere [33]. Since we were interested in testing any possible variation in GLS, regardless of its actual clinical impact, a quite restrictive non-inferiority limit (d) of 2 was selected. We calculated that in case no true difference would exist between the Post-VACCINE and control groups, 78 patients (39 per group) would have been required to be 90% sure that the lower limit of a one-sided 95% confidence interval (or equivalently a 90% two-sided confidence interval) would fall above the pre-specified non-inferiority limit. Sample size and power calculations were run using Sealed Envelope calculator (Sealed Envelope Ltd. (London, UK) 2012). A power calculator for continuous outcome non-inferiority trials is available online: https://www.sealedenvelope.com/power/continuous-noninferior/ (accessed on 14 March 2022). Graphs in the figures were drawn with Past software (version 4.02). A *p*-value of less than 0.05 was considered statistically significant in this study. 

## 3. Results

### 3.1. Study Population Characteristics

Between April and June 2022, we prospectively evaluated 39 consecutive children and adolescents who have been fully vaccinated for COVID-19 for at least 3 months.

Among the 39 post-VACCINE cases, 24 were females (61%) with a mean age of 12.6 ± 2.6 years at the date of cardiological assessment. 

Twenty-three (59%) patients received a two-dose primary series of mRNA vaccination; two (5%) children who were vaccinated within 12 months after a previous COVID-19 received only a one-dose primary series of mRNA vaccination; and 14 (36%) subjects received a two-dose primary series of mRNA vaccination plus a first booster dose.

Thirty-five (90%) individuals were previously healthy, without any evidence of previous cardiac disorders. The remaining four individuals presented with a bicuspid aortic valve with mild regurgitation and persistence of left superior vena cava (N = 1.3%), a mitral valve prolapse and mild regurgitation (N = 2.5%), and a bicuspid aortic valve and normal functioning (N = 1.3%). Thirty-three (84%) children practiced sports, of whom 8 (24%) and 25 (76%) individuals performed agonistic and non-agonistic activities, respectively.

Thirty-two (82%) individuals were naïve-vaccinated and then classified as post-VACCINE-only cases, while seven (18%) individuals were vaccinated after previous COVID-19 and were classified as post-VACCINE+COVID-19 cases. Among post-VACCINE+COVID-19 children, two and five individuals developed COVID-19 in the Parental and Omicron wave, respectively. Of these, four (57%) had an asymptomatic COVID-19, while the remaining three (43%) experienced mild COVID-19. No children had a moderate or severe disease that required hospitalization. Only one patient reported the persistence of symptoms (i.e., headache and depression) after COVID-19. A mean time of 6.4 months ±7 elapsed between the baseline of the infection and the cardiological assessment. 

Among the overall 39 post-VACCINE cases, 6 individuals (15%) experienced mild and self-limited adverse reactions during the 48 hours after COVID-19 vaccination: one patient (3%) encountered fatigue, two patients (5%) fever, one patient (3%) nausea and/or emesis, and three patients (8%) headache. No patients developed cardiologic symptoms, including chest pain. Moreover, no patients developed myocarditis or pericarditis in the following 3 months after vaccination.

The study cohort was compared with 39 (24 females—61%) age- and gender-matched healthy controls.

Baseline and standard echocardiographic characteristics of the post-VACCINE cases and CTRLs are presented in Table 1. As shown in Table 1, post-VACCINE individuals and CTRL were comparable regarding age, gender, and BSA. 

All the cardiac evaluations were performed after an average follow-up of 3.8 ± 1.5 months since the last received dose of the mRNA vaccine.

### 3.2. Standard Echocardiographic Measurements

Left ventricular dimensions (post-VACCINE LVEDD: 41.6 ± 5.2 mm; CTRL: 41.7 ± 5.7; *p* = 0.940, Figure 2A, Table 1) were comparable between the post-VACCINE and CTRL groups (post-VACCINE RVSP: 16.1 ± 4.0 mmHg vs. CTRL: 15.2 ± 3.1, *p* = 0.394, Figure 2C, Table 1). Furthermore, the right ventricular longitudinal function, assessed with the tricuspid annular plane systolic excursion method (TAPSE), and right ventricular systolic pressure (RVSP) were comparable between the two groups (post-VACCINE: 21.3 ± 2.9 mm; CTRL: 21.0 ± 2.4; *p* = 0.575) (Figure 2B,C, Table 1). Global systolic function, assessed by left ventricular ejection fraction, was also similar between the two groups (post-VACCINE LVEF: 64.1 ± 4.7%; CTRL: 64.4 ± 4.7%; *p* = 0.732) (Table 1, Figure 3A).

No signs of significant coronary artery dilation (Z scores all < 2) or pericardial effusion were appreciated among post-VACCINE cases (*p* = 0.471). 

Finally, as shown in Table 1, LVEDD, LVEF, RVSP, and TAPSE were found to be comparable in post-VACCINE children who had also experienced COVID-19 infection compared with post-VACCINE children without a previous diagnosis of COVID-19 (LVEDD post-VACCINE+COVID-19: 39.4 ± 3.3 mm; post-VACCINE-only: 42.1 ± 5.4 mm; *p* = 0.118; LVEF VACCINE+COVID-19: 62.4 ± 3.1%; post-VACCINE-only: 64.4 ± 4.9%; *p* = 0.190; RVSP post-VACCINE+COVID-19: 14.3 ± 1.1 mmHg; post-VACCINE-only: 16.3 ± 4.1 mmHg; *p* = 0.092; TAPSE VACCINE+COVID-19: 21.5 ± 2.8 mm; post-VACCINE-only: 21.2 ± 3.0 mm %; *p* = 0.791).

### 3.3. Left Ventricular Longitudinal Strain

The global longitudinal strain was in the normal range in all individuals, with no differences between children after COVID-19 vaccination and controls (post-VACCINE: −21.7 ± 2.3%; CTRL: 21.2 ± 1.8%; *p* = 0.338, Figure 3B). Of note, the 95% confidence interval of the post-VACCINE group (20.98–22.42) did not cross the pre-specified non-inferiority limit.

However, while LVEF was comparable (LVEF post-VACCINE+COVID-19: 62.4 ± 3.1%; post-VACCINE-only: 64.4 ± 4.9%; *p* = 0.190) (Figure 4A), GLS was found to be slightly but significantly reduced in post-VACCINE+COVID-19 cases compared to post-VACCINE-only cases (post-VACCINE+COVID-19: −19.9 ± 1.1%; post-VACCINE-only: −22.0 ± 2.3%; *p* = 0.002, Figure 4B). 

Moreover, to better evaluate the impact of age on the left ventricular longitudinal strain, we stratified post-VACCINE cases among two age classes (5–11 and 12–17 years old). Among the 15 patients aged 5–11 years and the 24 individuals aged 12–17 years, no differences were documented for LVEF (63.5 ± 4.8% vs. 64.5 ± 4.6%, *p*-value = 0.484), TAPSE (20.1 ± 2.9 mm vs. 21.5 ± 2.9 mm, *p*-value = 0.589) or for GLS (−21.9 ± 1.8% vs. −21.4 ± 2,6%, *p*-value = 0.513).

Similarly, among the post-VACCINE cases, we compared the cardiac measures between the group of six children (15%) who reported mild adverse reactions to vaccination and the 33 individuals who did not. No significant differences were observed between children with and without mild adverse events after vaccination with regard to GLS (−21.4 ± 1.8% vs. −23.1 ± 3.4, *p*-value = 0.266) and TAPSE (21.3 ± 2.9 mm vs. 21.2 ± 3.2 mm, *p*-value = 0.935).

Furthermore, those post-VACCINE cases who presented with cardiac comorbidities did not exhibit any significant difference compared with the remaining children of the group with regard to GLS (−21.7 ± 2.3 vs. 21.2 ± 2.7, *p*-value = 0.761)) and TAPSE (−21.4 ± 2.9 vs. 20.1 ± 2.8 *p*-value = 0.761).

### 3.4. Reproducibility Analysis

Indices of intra- and inter-observer variability were very good for longitudinal strains (7 ± 8%, ICC: 0,93 and 7 ± 8%, ICC: 0,92, respectively).

## 4. Discussion

To the best of our knowledge, this is the first study showing a comprehensive assessment of the cardiac function following COVID-19 vaccination and a detailed prospective analysis of the standard echocardiography and speckle-tracking echocardiography (STE) parameters in a cohort of healthy children who received COVID-19 vaccination, with or without prior SARS-CoV-2 infection.

The major results of this study reveal: (1) children who underwent COVID-19 vaccination did not exhibit an impairment of left or right ventricular function in terms of standard parameters and longitudinal strain; (2) global longitudinal strain, but not LVEF, was significantly reduced, albeit slightly, in children who had undergone COVID-19 infection and received COVID-19 vaccination, as compared to children who received COVID-19 vaccination but did not face COVID-19 infection. Although this difference remained statistically significant even after the Bonferroni correction for multiple comparisons, these results must be taken with caution as this was not the main study objective, not a pre-specified analysis. In addition, despite the fact that these findings are well in line with previous observations, post-hoc calculations revealed a low statistical power; (3) children who experience a mild adverse reaction after COVID-19 vaccination did not encounter any significant impairment of either left or right ventricular function.

Since the emergency use authorization, cases of myopericarditis following COVID-19 mRNA vaccination have been documented worldwide, particularly in young adults and adolescents [32,34]. Previous research has reported largely positive outcomes in adults with myocarditis following COVID-19 mRNA vaccination, including symptom resolution, cardiac function preservation, and no complications [35,36]. 

A recent systematic review and meta-analysis, from a broad range of populations, discovered a low incidence rate and predominantly positive early outcomes of myopericarditis associated with COVID-19 mRNA vaccination in young adults and adolescents [37].

However, information on the echocardiographic features after COVID-19 vaccination in adolescents and young adults is scarce, especially when compared to adults, and often includes small case series. 

Our results in a prospective cohort of healthy children documented the absence of early detrimental effects on the cardiac function of the COVID-19 vaccine. Not only standard echocardiographic parameters but also the more sensitive and advanced indices of longitudinal cardiac deformation excluded vaccine-associated cardiac damage in our cohort. Adverse reactions reported from our cohort have been scarce and of a mild entity. Moreover, even in these latter cases, we did not observe any significant cardiac reaction in terms of reduced ventricular function or pericardial effusion.

On the other hand, the results regarding the impairment of myocardial longitudinal strain after SARS-CoV2 infection were in line with previously published data [19,21,38] on both children and adult patients with an asymptomatic or mildly symptomatic COVID-19 course, demonstrating a slight but significant reduction in cardiac longitudinal deformation in children with a previous SARS-CoV-2 infection compared to those who were only vaccinated.

COVID-19 vaccines represent the more effective strategies to contain the ongoing COVID-19 pandemic, preventing the SARS-CoV-2 transmission, and are crucial to protect children from hospitalization, long-COVID syndrome, MIS-C, and death. These findings may contribute to an improved understanding of the vaccine effect on the heart of children and young adults and wish to support informed decision-making for parents. 

However, our study has several limitations. Firstly, the study was limited by a small sample size, which included only a limited number of children with prior COVID-19 infection. As a result, it was not possible to fully assess the impact of infection and vaccination on cardiac impairment or to determine the contribution of each viral variant to ventricular alterations. Furthermore, the study’s punctual echocardiographic evaluation and lack of long-term cardiological follow-up prevented the assessment of the reversibility of cardiac impairment over time following vaccination. Moreover, the age distribution of the patients in this study may have affected the statistical analysis. The study did not use a fixed number of individuals per age group to minimize selection biases, which means that some age groups may have been smaller than others. Therefore, while data for all age groups are reported, this information should be considered exploratory rather than definitive. Additionally, the data for some age subgroups may not be robust enough due to the limited number of individuals included in the analysis. In addition, the measurement of longitudinal strain using speckle-tracking analysis has certain limitations that are commonly associated with this technique. These limitations are primarily related to the quality of the images obtained, which can be affected by factors such as poor compliance in younger patients, a high heart rate, and lung artifacts. While acoustic windows tend to be more echogenic in younger patients, these factors can still impact image quality. Furthermore, the use of a pre-enrolled historical control group introduced the potential for measurement bias, as echocardiographic scans were performed by different operators. However, speckle-tracking analysis measurements were performed by the same two expert echocardiographers on pre-recorded clips from both the vaccinated and control children, with optimal ICCs (see Results Section 3). However, the use of a pre-pandemic cohort helped to minimize the risk of enrolling patients with unknown COVID-19 diseases.

## 5. Conclusions

In conclusion, we did not observe an impairment in global longitudinal strain signaling any cardiac impairment after mRNA vaccination in a small cohort of both children and adolescents. Further studies conducted on larger samples of children and adolescents who underwent mRNA vaccination are needed to confirm our findings. In addition, active monitoring and research are necessary to better comprehend and prevent SARS-CoV-2-associated cardiac damage in the pediatric population.

## Figures and Tables

**Figure 1 vaccines-11-01348-f001:**
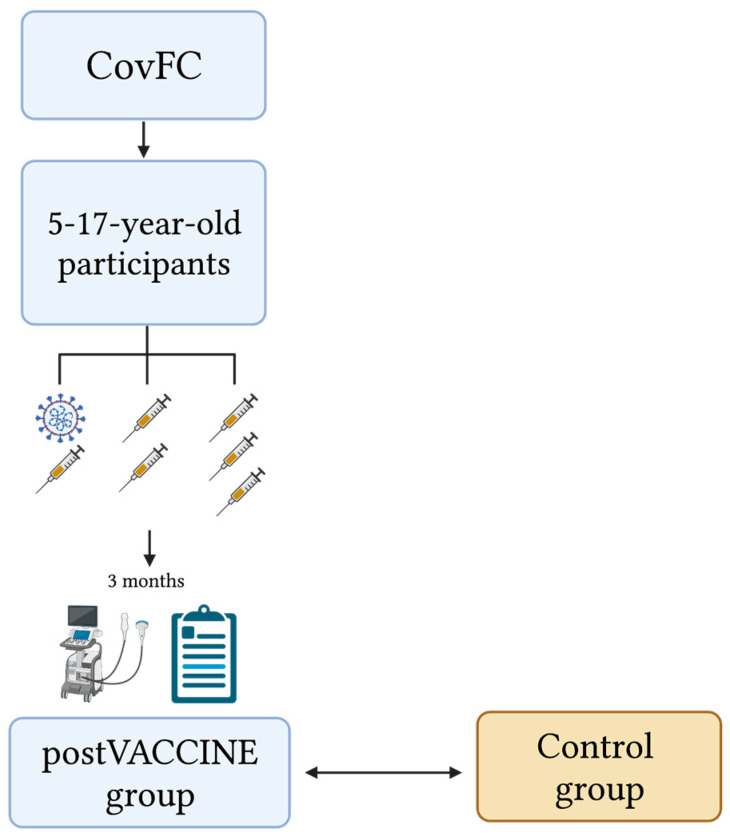
Patients’ enrollment and study design flow-chart. The post-VACCINE group included participants aged 5–17 years who were recruited from the COVID-19 Family Cluster Follow-up Clinic (CovFC) and received: (i) one dose of the Comirnaty-BioNTech/Pfizer or Spikevax vaccine within 12 months after a previous laboratory-confirmed COVID-19; (ii) two doses of the Comirnaty-bioNTech/Pfizer or Spikevax vaccine; or (iii) two doses of the Comirnaty-bioNTech/Pfizer or Spikevax vaccine plus a booster dose. The post-VACCINE group underwent cardiac assessment 3 months post-vaccination. The control group included age- and gender-matched healthy children who underwent cardiac assessment before the onset of the COVID-19 pandemic.

**Figure 2 vaccines-11-01348-f002:**
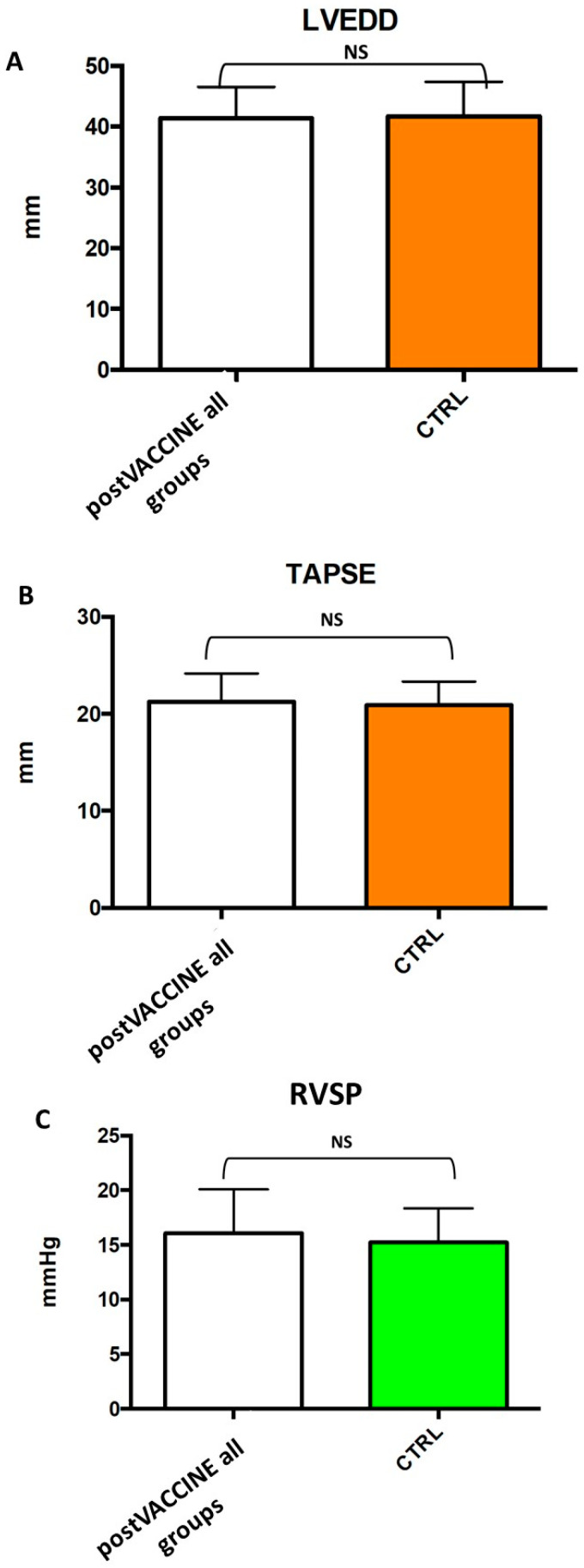
Comparison of standard echocardiographic measurements between post-VACCINE cases and controls (CTLR). The bar graphs show that LVEDD (**A**), TAPSE (**B**) and RVSP (**C**) were comparable between the post-VACCINE and CTRL groups with a *p* value = NS. LVEDD: left ventricular end-diastolic diameter; RVSP: right ventricular systolic pressure; TAPSE: tricuspid annular plane systolic excursion.

**Figure 3 vaccines-11-01348-f003:**
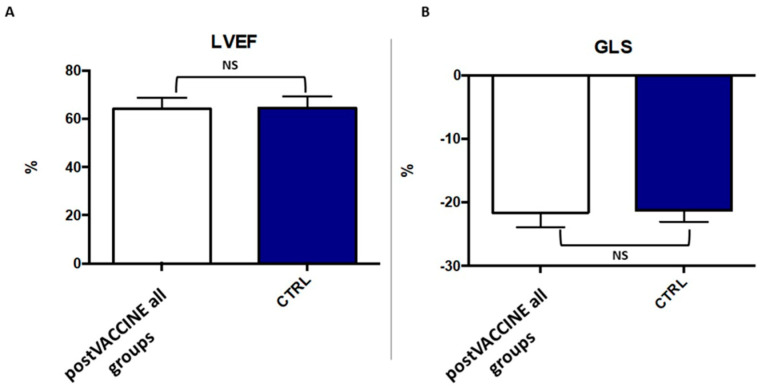
Comparison of left ventricular longitudinal strain between post-VACCINE cases and controls (CTRL). The figure illustrates similar values of LVEF (**A**) and GLS (**B**) in post-VACCINE and CTRL groups with a *p*-value of 0.732 and 0.338, respectively. LVEF: left ventricular ejection fraction; GLS: global longitudinal strain.

**Figure 4 vaccines-11-01348-f004:**
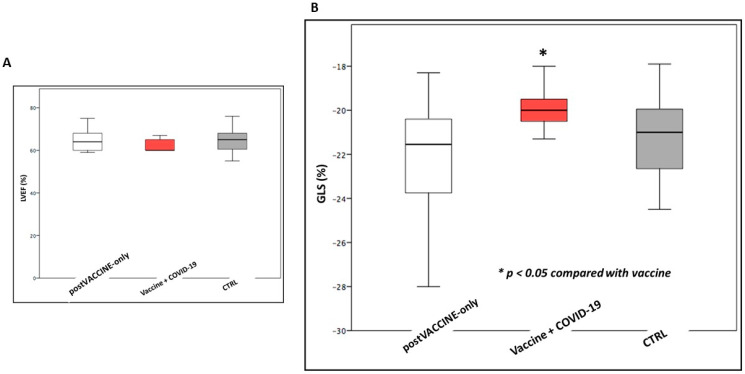
Comparison of global systolic function among vaccinated individuals with (post-VACCINE+COVID-19) or without (post-VACCINE-only) previous COVID-19 and controls (CTRL). Panel (**A**) and (**B**) show how LVEF and GLS were slightly but significantly reduced in post-VACCINE children who also experienced COVID-19, compared with post-VACCINE children without a previous COVID-19 (GLS: post-COVID-19+VACCINE: −19.9 ± 1.1%; post-VACCINE-only: −22.0 ± 2.3%; *p* = 0.002). LVEF: left ventricular ejection fraction; GLS: global longitudinal strain.

**Table 1 vaccines-11-01348-t001:** Demographic and echocardiographic characteristics of the study population.

	CTRL (n = 39)	post-VACCINE (All Groups) (n = 39)	post-VACCINE+ COVID-19 (n = 7)	*p* Value between CTRL (n = 39) and post-VACCINE (n = 39) Groups
Age (years)	12.7 ± 3.1	12.6 ± 2.6	12.7 ± 3.2	0.937
Female. n (%)	24 (61%)	24 (61%)	7 (100%)	0.817
BSA (m^2^)	1.41 ± 0.27	1.36 ± 0.24	1.27 ± 0.27	0.319
LVEDD	41.7 ± 5.7	41.6 ± 5.2	39.4 ± 3.3	0.940
LVEDD Z score	−0.7 ± 1.0	−0.5 ± 1.0	−0.8 ± 0.7	0.467
LVESD (mm)	26.1 ± 4.2	26.1 ± 3.5	24.6 ± 3.5	0.999
LVESD Z score	−0.4 ± 1.0	−0.3 ± 0.9	−0.6 ± 0.9	0.645
IVSd (mm)	7.3 ± 1.3	7.5 ± 1.6	7.5 ± 1.2	0.624
IVSd Z score	+0.3 ± 0.8	+0.4 ± 0.9	+0.6 ± 0.4	0.695
PWd (mm)	6.9 ± 1.1	7.0 ± 1.4	7.1 ± 1.4	0.626
PWd Z score	+0.5 ± 0.6	+0.6 ± 0.9	+0.9 ± 0.9	0.780
RVSP (mmHg)	15.2 ± 3.1	16.1 ± 4.0	14.3 ± 1.1	0.394
TAPSE (mm)	21.0 ± 2.4	21.3 ± 2.9	21.5 ± 2.8	0.575
LVEF (%)	64.4 ± 4.7	64.1 ± 4.7	62.4 ± 3.1	0.732
GLS (%)	−21.2 ± 1.8	−21.7 ± 2.3	−19.9 ± 1.1	0.338

BSA: Body surface area; LVEDD: Left ventricular end-diastolic diameter; LVESD: Left ventricular end-systolic diameter; IVSd: Interventricular septum in diastole; PWd: Posterior wall in diastole; RVSP: Right ventricular systolic pressure; TAPSE: Tricuspid annular plane systolic excursion; LVEF: Left ventricular ejection fraction; GLS: Global longitudinal strain.

## Data Availability

Data are available from the corresponding author on reasonable request.

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
