# Peer review of "Cardiac Function Evaluation after SARS-CoV-2 mRNA Vaccination in Children and Adolescents: A Prospective Speckle-Tracking Echocardiography Study"

_vaccines, 2023, doi:10.3390/vaccines11081348_

Round 1
Reviewer 1 Report
This paper reports a descriptive statistics study that appears to follow methodological protocols for such studies and to be well done.
The main suggestion that I have for revision of the paper is that the authors should include more elaborate verbal descriptions of the contents of the tables and figures in the text. For instance, the description of the contents of Table 1 in the text is "Baseline and standard echocardiographic characteristics of the postVACCINE cases and CTRLs are presented in Table 1." In addition to this statement, you need to comment on the contents of the table and any statistical differences between the group of note. Another example, the contents of some of the figure are shown in box A and box B, but these are not carefully identified and commented upon in the text. Yes, with a careful reading of the paper, these things can be understood by readers. But you should go ahead and state them clearly in the text.
Reviewer 2 Report
It is a single-center, prospective, case-control study enrolling children and adolescents. Three months after receiving the primary mRNA vaccination or booster dose, the patients underwent a cardiac assessment, including standard echocardiography and speckle-tracking echocardiography.
The Authors showed that mRNA vaccination was not associated with alterations in cardiac structures, systolic functions, and LV deformations measures
It is a quite interesting manuscript. The topic of this manuscript falls within the scope of Vaccines. The topic is relevant, interesting, and original.
The data has been provided with vigorous statistical analysis. The Authors have presented sufficient data. The appropriate table and figures have been provided. The manuscript is well written. The article is easy to read and logically structured. The methods are adequately described. The Authors also added very good limitations. The conclusions are consistent with the presented evidence and arguments. They address the main question posed.
There is only one comment in the reviewer's opinion which should be taken under consideration by the Authors:
1. Please include in the section material and methods -study design diagram
Reviewer 3 Report
In the article titled "Cardiac Function Evaluation after SARS-CoV-2 mRNA Vaccination in Children and Adolescents: A Prospective Speckle-Tracking Echocardiography Study" by Jolanda Sabatino and colleagues. The study included 39 post-vaccine cases. Ninety-two percent (N=36) of the patients were previously healthy. There were no differences between cases and controls in left ventricular diameters, left ventricular ejection fraction (LVEF), and tricuspid annular plane systolic excursion (TAPSE). The global longitudinal strain (GLS) was within the normal range in all individuals, with no differences between post-vaccine cases and controls (-21.7±2.3% vs 21.2±1.8%; p=0.338). The GLS was, however, significantly reduced in post-VACCINE children who had previously received a COVID-19 against nave individuals with a COVID-19 (postCOVID-33 + 19 + VACCINE: -19.9 - 1.2% vs only post-VACCINE: -22.0 - 2.3%; p 0.001). In regard to the present manuscript, I would like to make a few comments.
-The introduction and material and methods sections appear well written
-There is a major problem in the current study with the sample recruited, as it is a very small sample for any statistical validation, and the statistical power may be insufficient
-The sample size plays a crucial role in this kind of study because the difference found could be related to another variable not measured. With a large sample, these kinds of problems are minimized.
Round 2
Reviewer 3 Report
Thank you for attending to my previous comments in the first revision round. Nevertheless, I believe that more subjects are required to complete your study
Round 3
Reviewer 3 Report
Thank you for adding more detailed information to the statistical analyses. Having read the article with interest, I do not need to make any further comments